# Dynamic Evidence Decoupling for Trusted Multi-view Learning

## ABSTRACT

Multi-view learning methods often focus on improving decision accuracy, while neglecting the decision uncertainty, limiting their suitability for safety-critical applications. To mitigate this, researchers propose trusted multi-view learning methods that estimate classification probabilities and uncertainty by learning the class distributions for each instance. However, these methods assume that the data from each view can effectively differentiate all categories, ignoring the semantic vagueness phenomenon in real-world multi-view data. Our findings demonstrate that this phenomenon significantly suppresses the learning of view-specific evidence in existing methods. We propose a Consistent and Complementary-aware trusted Multi-view Learning (CCML) method to solve this problem. We first construct view opinions using evidential deep neural networks, which consist of belief mass vectors and uncertainty estimates. Next, we dynamically decouple the consistent and complementary evidence. The consistent evidence is derived from the shared portions across all views, while the complementary evidence is obtained by averaging the differing portions across all views. We ensure that the opinion constructed from the consistent evidence strictly aligns with the ground-truth category. For the opinion constructed from the complementary evidence, we only require it to reflect the probability of the true category, allowing for potential vagueness in the evidence. We compare CCML with state-of-the-art baselines on one synthetic and six real-world datasets. The results validate the effectiveness of the dynamic evidence decoupling strategy and show that CCML significantly outperforms baselines on accuracy and reliability. We promise to release the code and all datasets on GitHub and show the link here.

## CCS CONCEPTS

• **Computing methodologies → Machine learning**.

## KEYWORDS

Trusted Multi-view Learning, Uncertainty-aware Deep Learning, Dynamic Multi-view Learning.

## 1 INTRODUCTION

In real-world scenarios, different data modalities or features could be treated as multiple views. For example, in autonomous vehicle systems, cameras and lidars collect images and points; in the field of healthcare, a patient's comprehensive condition can be assessed through multiple types of examinations. Multi-view learning

**Unpublished working draft. Not for distribution.**

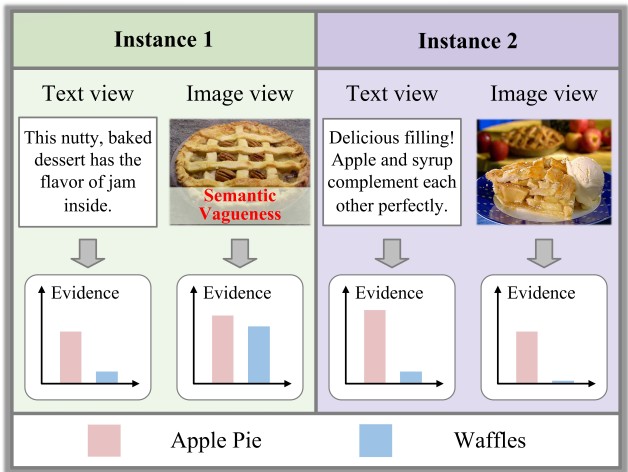

**Figure 1: Visualization of the dynamic semantic vagueness phenomenon. The ground-truth category of the first instance is "apple pie". When considering the image view alone, it becomes challenging to differentiate between the categories "waffles" and "apple pie". For the second instance, both views provide explicit differentiation between the categories.**

aims to synthesize both consistent and complementary information from these multiple views, leading to a more comprehensive understanding of the data [4, 20, 32]. It has generated significant and wide-ranging influence across multiple research areas, including classification [17], clustering [13, 29], retrieval [41] and large language models [22].

Most existing multi-view learning methods primarily emphasize enhancing decision accuracy, often overlooking the crucial aspect of decision uncertainty. This limitation significantly restricts the applicability of multi-view learning in safety-critical scenarios, such as autonomous vehicle systems. Therefore, to predict the reliability of the results, researchers have proposed many multi-view uncertainty quantification methods in recent years. The pioneering work [9], Trusted Multi-view Classification (TMC), calculates and aggregates the evidence [27] of all views. TMC utilizes this aggregated evidence to parameterize the class distributions, enabling the estimation of class probabilities and uncertainties. In order to train the model effectively, TMC requires the estimated class probabilities to align with the ground-truth labels. Building upon this research, researchers have proposed novel evidence aggregation paradigms such as sum [21], (weighted) average [35], element-wise dot product [10], etc. These methods enhance the reliability and robustness in the presence of various challenges, such as feature noise [8, 25, 42], incomplete views [33], etc.

Regretfully, the evidence aggregation paradigms in these methods rely on an assumption: the data of each view can distinguish all categories. However, real-world multi-view data exhibit the *semantic vagueness phenomenon*, which means that one view may

exhibit ambiguity or uncertainty in its categorization. For example, as shown in Figure 1, the ground-truth category of the first instance is "apple pie", while the image view is difficult to differentiate between the categories "waffles" and "apple pie". For this view, existing evidence aggregation paradigms encourage only the evidence of the category "apple pie" is large due to the common evidence complying with this. This overlooks the fact that the evidence for the category "waffles" is also substantial in the data, which would significantly impact the overall learning process and compromise its effectiveness. This motivates us to delve into the semantic vagueness problem in trusted multi-view learning.

In this paper, we propose a Consistent and Complementary-aware trusted Multi-view Learning (CCML) for this problem. First, we construct view-specific evidential Deep Neural Networks (DNNs) to learn the view-specific evidence, which represents the level of support for each category obtained from the data. In the multi-view fusion stage, we dynamically decouple the consistent and complementary evidence. The consistent evidence is derived from the consistent portions across all views, while the complementary evidence is obtained by averaging the differing portions from all views. This separation allows us to capture both the shared information and the unique aspects of each view. In the training stage, we enforce strict alignment between the opinion constructed from the consistent evidence and the ground-truth category. This is achieved by adjusting the probabilities of the true and false categories, as well as enhancing the separation between them. As for the opinion constructed from the complementary evidence, we only require it to reflect the probability of the true category, allowing for potential vagueness in the evidence. In the test stage, we aggregate the consistent and complementary evidence to make a decision.

The main contributions of this work are summarized as follows: 1) we identify the semantic vagueness phenomenon in multi-view data, which can significantly suppress the learning of view-specific evidence in existing trusted multi-view learning methods; 2) we propose the CCML method to tackle this problem. CCML effectively addresses the negative impact of semantic vagueness through two key strategies: dynamically decoupling the consistent and complementary evidence, and allowing potential vagueness in the complementary evidence; 3) we conduct empirical comparisons between CCML and state-of-the-art trusted multi-view learning baselines on a synthetic toy dataset and six real-world datasets. The experimental results not only validate the effectiveness of the proposed dynamic decoupling strategy but also demonstrate that CCML surpasses the baseline methods in terms of accuracy and reliability.

## 2 RELATED WORK

The proposed CCML is a new uncertainty-aware multi-view fusion methods. Therefore, in this section, we briefly review two lines of related work, multi-view fusion and uncertainty-aware deep learning, to better motivate this work.

### 2.1 Multi-view Fusion

Multi-view fusion is highly effective in a wide range of tasks, as it combines information from multiple sources or modalities [6, 12, 39]. Based on the fusion strategy employed, existing deep multi-view fusion methods can be broadly categorized into two

main pipelines: feature fusion [1, 19] and decision fusion [14]. Feature fusion methods aim to capture the interactions between different views at the feature level. For example, canonical correlation analysis and its variants [1, 40] maximize the correlation of the multi-view latent representations. Matrix factorization methods [34] decode the multi-view common representation to view-specific data via basis matrices. Following this line, Xu *et al.* explicitly model consistent and complementary information [2] at the highest abstraction level by the group sparseness constraint [34]. More recently, researchers have used deep neural networks to decouple the complex consistent and complementary information at representation level [4, 37]. A major challenge of feature fusion methods is that low-quality views may adversely affect the representation of other views. Decision fusion methods [5, 23, 26, 28] mitigate this problem by integrating the decision results from different views. We follow this line and propose a new method for decoupling consistent and complementary information at the decision level.

### 2.2 Uncertainty-aware Deep Learning

Traditional deep learning methods have made remarkable progress in various domains. However, they are unable to provide uncertainty estimates in predictions, which is increasingly important in real-world scenarios. To tackle this challenge, researchers have proposed uncertainty-aware deep learning methods. One approach is Bayesian Neural Networks [24], which considers the variation in results caused by variations in the data distribution as uncertainty. However, the high computational cost associated with Bayesian neural networks limits their practical applications. As a result, researchers begin to explore more efficient methods for estimating uncertainty. The Monte Carlo dropout method [7] is one such approach. It involves using multiple instances of input data to obtain multiple prediction results, from which uncertainty measures can be calculated. Another method, EDL [27], calculates category-specific evidence and considers the lack of evidence as a source of uncertainty using a single deep neural network. Recently, TMC [9] extends EDL to the multi-view learning area. Following this line, researchers have proposed various evidence aggregation paradigms, including sum [21], (weighted) average [35], element-wise dot product [10], etc. However, these paradigms often overlook the phenomenon of semantic vagueness present in real-world multi-view data, which can significantly hinder the learning of view-specific evidence. The proposed CCML effectively addresses the issue by allowing for potential vagueness in the complementary evidence.

## 3 THE METHOD

In this section, we first introduce the trusted multi-view classification problem and semantic vagueness phenomenon, then present the pipeline and loss function of CCML in detail.

### 3.1 Notations and Problem Definition

In this section, we introduce the trusted multi-view classification problem and semantic vagueness phenomenon in detail. For the $C$ classification problem, considering a dataset $D = \{\{x_n^v\}_{v=1}^{V}, y_n\}_{n=1}^{N}$ has $N$ instances with $V$ views, where $x_n^v$ denotes the feature vector for the $v$-th view of the $n$-th instance, and the one-hot vector $y_n \in \{0, 1\}^C$ denotes the ground-truth label of the $n$-th instance.

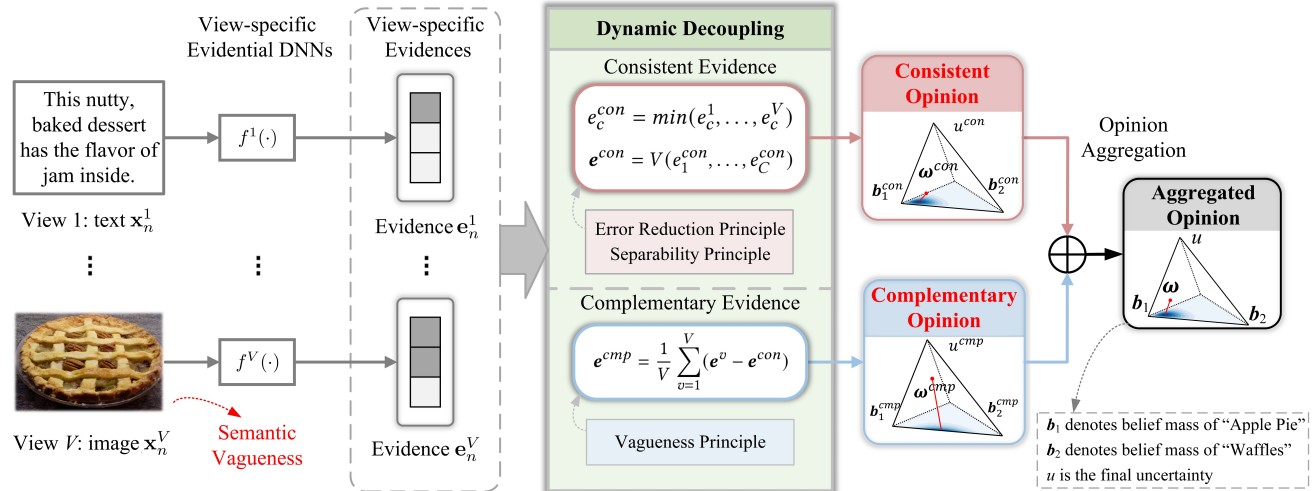

**Figure 2: Illustration of CCML. CCML initially constructs view-specific evidential DNNs to acquire the view-specific evidence and subsequently dynamically decouples the consistent and complementary evidence. During training, CCML ensures precise alignment between the opinion constructed from the consistent evidence and the ground-truth category. Regarding the complementary evidence, CCML only necessitates reflecting the probability of the true category, accommodating potential evidence vagueness. During testing, CCML combines consistent and complementary evidence to reach a decision.**

The semantic vagueness phenomenon indicates that some elements of $\{x_n^v\}$ may not well distinguish certain categories. For example, as shown in Figure 1, the image view of the first instance is hard to differentiate between the categories "waffles" and "apple pie". The goal is to accurately predict $y_n$ and provide the associated prediction uncertainties $u_n^v \in [0, 1]$ to measure the decision reliability.

## 3.2 CCML Pipeline

As shown in Figure 2, the overall architecture is a decision-level fusion pipeline, which consists of the view-specific evidence learning stage and the evidential multi-view fusion stage. The view-specific DNNs $\{f^v(\cdot)\}_{v=1}^V$ learn the view-specific evidence, which indicates the level of support for each category based on the data. In the fusion stage, we dynamically decouple the consistent evidence ($e^{con}$) and complementary evidence ($e^{cmp}$). $e^{con}$ is extracted from the consistent portions across all views, while $e^{cmp}$ is obtained by averaging the differing portions across all views. In the training stage, we establish different principles for $e^{con}$ and $e^{cmp}$, respectively. For the testing stage, we aggregate $e^{con}$ and $e^{cmp}$ to make a decision.

*3.2.1 View-specific Evidence Learning.* Many conventional multi-view learning methods utilize softmax layers to produce standard outputs to address multi-classification problems in neural classification networks. However, these methods only provide class probabilities without considering the reliability of the results [11]. Due to the single-point estimation paradigm of softmax scores, they tend to produce overconfident outputs, even when the predictions are incorrect. To address these issues and achieve accurate uncertainty prediction, we employ EDL [27] by replacing the softmax layer with a ReLU activation to obtain non-negative evidence.

We also introduce the subjective logic [15] framework to form opinions. In this framework, the parameter $\boldsymbol{\alpha}$ of the Dirichlet distribution $Dir(\boldsymbol{p}|\boldsymbol{\alpha})$ is associated with the belief distribution in the framework of evidence theory, where $\boldsymbol{p}$ is a simplex representing the probability of class assignment. We collect evidence $\{e_n^v\}$, using view-specific evidential DNNs $\{f^v(\cdot)\}_{v=1}^V$. The corresponding Dirichlet distribution parameters are $\boldsymbol{\alpha}^v = e^v + 1 = [\alpha_1^v, \cdots, \alpha_C^v]^T$. After obtaining the distribution parameters, we can calculate the subjective opinion, $\boldsymbol{\omega}^v = (\boldsymbol{b}^v, u^v)$ of the view including the quality of beliefs $\boldsymbol{b}^v$ and the quality measure of uncertainty $u$, where $\boldsymbol{b}^v = (\boldsymbol{\alpha}^v - 1)/S^v = e^v/S^v$, $u^v = K/S^v$, and $S^v = \sum_{k=1}^K \alpha_k^v$ is the Dirichlet intensity.

After training evidential DNNs to learn view-specific evidence, we observe that the evidence produced by vagueness views contains support for multiple categories. Among these categories, the true categories would generate relatively higher amounts of evidence than other categories. Therefore, we cannot require only the ground-truth category has large evidence. We utilize this property in the subsequent evidence fusion stage.

*3.2.2 Consistent and Complementary-aware Multi-view Fusion.* Existing trusted multi-view learning methods face limitations when addressing the phenomenon of semantic vagueness. This is primarily because, during the generation of view-specific evidence, the indistinguishable error categories in vague views produce a similar amount of evidence as the indistinguishable categories. Many methods struggle to distinguish between equal amounts of evidence for correct and incorrect categories. As a result, we are motivated to tackle this problem by decoupling the consistent and complementary evidence and allowing for potential vagueness in the complementary evidence.

For views $v_i$ and $v_j$, the view-specific deep learning network of view $v_i$ will produce similar amounts of evidence for classes $c_1$ and $c_2$, while the network of view $v_j$ will produce similar amounts of evidence for classes $c_2$ and $c_3$ when classifying instances of class $c_2$, Effective classification should be achieved through complementary information from view $v_i$ and $v_j$ respectively ($v_i$ determines that the instance is one of $c_1$ or $c_2$, and $v_j$ determines that the instance is one of $c_2$ or $c_3$, the final classification result should be $c_2$ utilizing the complementary information of $v_i$ and $v_j$).

Therefore, we need to decouple view-specific evidence of different views into consistent and complementary evidence. One previous work [36] addresses the semantic vagueness phenomenon by setting fixed relation degradation layers for semantic vagueness categories. However, the extent of semantic vagueness varies among different instances, as reflected by the difference in consistency and complementarity. It is necessary to use dynamic decoupling strategies according to the differences of consistency and complementarity in the classification tasks, instead of using fixed relations. For this purpose, we propose a dynamic consistent and complementary evidence decoupling strategy.

*Definition 3.1.* (Consistent and Complementary Evidences). For the view-specific evidences $\{e^v = (e_1^v, \ldots, e_C^v)\}_{v=1}^V$. The consistent evidence $e^{con}$ and complementary evidence $e^{cmp}$ are defined as:

$$e^{con} = V(e_1^{con}, \ldots, e_C^{con}), \tag{1a}$$

$$e_c^{con} = min(e_c^1, \ldots, e_c^V), c = 1, \ldots, C, \tag{1b}$$

$$e^{cmp} = \frac{1}{V} \sum_{v=1}^V (e^v - e^{con}), \tag{1c}$$

where $e^{con}$ denotes the minimum evidence of each class supported by each view. $e^{cmp}$ denotes the complementary evidence.

The consistent evidence $e^{con}$ is obtained by aggregating the consistent portions from all views, while the complementary evidence $e^{cmp}$ is calculated as the average of the differing portions across all views. This distinction is made because the complementary evidence of all views $\{e^v - e^{con}\}_{v=1}^V$ capture the varying information between different views, encompassing both complementarity and conflict. It is important to note that this information does not always enhance the accuracy of results; instead, it can introduce additional uncertainty. Consequently, retaining the entire set of complementary evidence is not advisable. Instead, a more suitable approach is to preserve it by taking the average [35]. By employing this strategy, we dynamically separate the consistent and complementary evidence from the view-specific evidence. Finally, we aggregate the two pieces of evidence on average to form the final opinion, which is used in the test stage. In the following section, we will outline how our deep learning networks are trained using distinct principles for handling consistent and complementary evidence.

## 3.3 Loss Function

In this section, we will present the training process of CCML. The objective is to deal with view-specific, consistent, and complementary evidence separately. Specifically, the consistent evidence should closely approximate the true label, while the complementary evidence serves as a supplementary component, allowing for more relaxed constraints. We will elaborate on these components below.

*3.3.1 View-specific Loss Function.* The evidential DNNs are obtained by converting the softmax layer of traditional DNNs into ReLU. Therefore, we obtain the non-negative outputs as evidence. We introduce an adjusted cross-entropy loss function to ensure that all views can generate appropriate non-negative view-specific evidence for classification:

$$\mathcal{L}_{ace}(\boldsymbol{\alpha}_n) = \int \left[ \sum_{j=1}^C -y_{nj} log p_{nj} \right] \frac{1}{B(\boldsymbol{\alpha}_n)} \sum_{j=1}^C p_{nj}^{\alpha_{nj}-1} d\boldsymbol{p}_n$$

$$= \sum_{j=1}^C y_{nj}(\psi(S_n) - \psi(\alpha_{nj})), \tag{2}$$

where $\psi(\cdot)$ is the digamma function. The view-specific loss function of $\boldsymbol{x}_n^v$ is defined as:

$$\mathcal{L}_{vs}(\boldsymbol{\alpha}_n^v) = \mathcal{L}_{ace}(\boldsymbol{\alpha}_n^v), \tag{3}$$

where $\boldsymbol{\alpha}_n^v = \boldsymbol{e}_n^v + 1$ is the parameters of the corresponding Dirichlet distribution, $\boldsymbol{e}_n^v = f^v(\boldsymbol{x}_n^v)$ represent the evidence vector predicted by the network.

*3.3.2 Consistent Loss Function.* The consistent evidence is obtained by aggregating the consistent portions of all views. Consequently, we impose a stringent alignment between the opinion constructed from the consistent evidence and the ground-truth category. To accomplish this, we adjust the probabilities assigned to the true and false classes while simultaneously enhancing the distinction between them. To achieve this objective, we introduce two principles: the Error Reduction Principle and the Separability Principle.

*Error Reduction Principle.* The error reduction principle highlights that during the training process, the evidence generated for incorrect categories may inadvertently increase due to the limited availability of counterexamples. This misleading evidence has the potential to introduce challenges to the classification process. Therefore, to reduce evidence for incorrect labels, we introduce the Kullback-Leibler (KL) divergence into the loss function:

$$\mathcal{L}_{KL}(\boldsymbol{\alpha}_n) = \lambda_t KL[D(\boldsymbol{p}_n|\widetilde{\boldsymbol{\alpha}}_n) \, || \, D(\boldsymbol{p}_n|\boldsymbol{1})]$$

$$= log\left( \frac{\Gamma(\sum_{c=1}^C \widetilde{\alpha}_{nc})}{\Gamma(K) \prod_{c=1}^C \Gamma(\widetilde{\alpha}_{nc})} \right)$$

$$+ \sum_{c=1}^C (\widetilde{\alpha}_{nc} - 1) \left[ \psi(\widetilde{\alpha}_{nc}) - \psi(\sum_{j=1}^C \widetilde{\alpha}_{nj}) \right], \tag{4}$$

where $D(\boldsymbol{p}_n|\boldsymbol{1})$ is the uniform Dirichlet distribution, $\widetilde{\boldsymbol{\alpha}}_n = \boldsymbol{y}_n + (1 - \boldsymbol{y}_n) \odot \boldsymbol{\alpha}_n$ is the Dirichlet distribution parameter after removing the evidence of the ground-truth category from the original parameter $\boldsymbol{\alpha}_n$. $\lambda_t = min(1, t/T) \in [0, 1]$ is annealing coefficient, acting as the balance factor. As the training process progresses, $\lambda_t$ continues to increase, enhancing the influence of KL divergence, to prevent premature convergence of misclassified instances to the uniform distribution.

*Separability Principle.* The Separability Principle emphasizes the importance of creating a significant distinction between the evidence supporting different categories during the classification process. This principle allows the classifier to more accurately distinguish between different categories. For instance, consider the

belief masses of opinions $b^1 = (0.4, 0.4)$ and $b^2 = (0.3, 0)$. The total amount of belief mass in $b^2$ is more than $b^1$. However, the greater degree of separation of $b^2$ makes it more contribution to the classification result than $b^1$. To enhance classification performance, we need to increase the degree of separation during the training process. Therefore, we quantify the degree of separation of the belief masses supporting different classes in opinion using the separation degree.

*Definition 3.2.* (Separation Degree). For the subjective opinion $\omega = (b, u, a)$, where $b = (b_1, \ldots, b_C)$, the separation degree can be defined as:

$$SD(b) = \sum_{i=1}^{C} \sum_{i \neq j}^{C} |b_i - b_j|. \tag{5}$$

Our goal is to increase the separation degree of consistent opinions. There are two approaches to increase it: the first approach involves adding a constraint to maximize the degree of separation. However, this approach may lead to an increase in the total amount of evidence across all categories. This is contrary to our original intention of controlling uncertainty based on the total amount of evidence. Therefore, we design the following method.

We first convert consistent evidence into consistent opinions $\omega^{con} = (b^{con}, u^{con}, a^{con})$ and adjust it to obtain the final consistent opinion $\widetilde{\omega^{con}} = (\widetilde{b^{con}}, u^{con}, a^{con})$:

$$\widetilde{b^{con}} = \begin{cases} \frac{\sum_{c=1}^{C} \widetilde{b_c^{con}}}{\sum_{c=1}^{C} b_c^{con}} (\widetilde{b_1^{con}}, \ldots, \widetilde{b_C^{con}}), & \sum_{c=1}^{C} b_c^{con} \neq 0, \\ 0, & otherwise, \end{cases} \tag{6}$$

where $\widetilde{b_c^{con}} = pow(b_c^{con}, \beta) \in [0, 1]$ is the belief mass supporting classes $c$, $\beta$ is a hyper-parameter which is bigger than 1. By the power operation $pow(b_c^{con}, \beta)$, the separation degree would increase. The reason behind this is the increasing disparity in the confidence mass that supports each category. We also demonstrate theoretically that this approach can increase the separation degree, which is elaborated in the appendix. Therefore, we only need this simple operation to achieve the increase of separation degree without changing the total amount of evidence.

Under the guidance of the above two principles, we can derive the consistent loss function:

$$\mathcal{L}_{con}(\alpha_n^{con}) = \mathcal{L}_{ace}(\widetilde{\alpha^{con}}) + \eta \mathcal{L}_{KL}(\widetilde{\alpha^{con}}). \tag{7}$$

where $\widetilde{\alpha^{con}}$ represents the corresponding parameters of the final Dirichlet distribution of the consistent opinion. The consistent loss function can refine the consistent evidence. After increasing separation degree, the cross-entropy loss function minimizes the model's deviation from the true label, and the KL loss function reduces the evidence for incorrect categories. The combination of these two loss functions can maximize the impact of consistent evidence on the model training process.

### 3.3.3 Complementary Loss Function.

*Vagueness Principle.* Complementary evidence usually represents complementary or even conflicting information between different views. Its reliability is generally lower than that of the consistent evidence. As a supplement to the consistent evidence, the complementary evidence need not necessarily enhance the degree of separation or reduce false labels. It accounts for potential vagueness in multi-view data. Therefore, it is only required to reflect the probability of the true category.

According to this principle, we define the complementary loss function as follows:

$$\mathcal{L}_{cmp}(\alpha_n^{cmp}) = \sum_{j=1}^{C} y_{nj}(\psi(S_n^{cmp}) - \psi(\alpha_{nj}^{cmp})), \tag{8}$$

where $\alpha_n^{cmp} = e_n^{cmp} + 1$ is the corresponding Dirichlet parameter of the complementary evidence.

### 3.3.4 Joint Loss.
By synthesizing the above objectives, the overall loss function for a specific instance $\{x_n^v\}_{v=1}^V$ is formulated as:

$$\mathcal{L} = \sum_{v=1}^{V} \mathcal{L}_{vs}(\alpha_n^v) + \delta \mathcal{L}_{con}(\alpha_n^{con}) + \gamma \mathcal{L}_{cmp}(\alpha_n^{cmp}), \tag{9}$$

where $\delta, \gamma > 0$ are hyper-parameters.

## 4 EXPERIMENT

In this section, we show the empirical results of CCML in making trusted decisions for multi-view inputs. We first apply CCML to a synthetic toy example to investigate its performance in solving the semantic vagueness problem, then we evaluate CCML on six real-world multi-view datasets and compare it with existing multi-view learning methods.

### 4.1 A Toy Example

The major advantage of CCML compared with pioneer uncertainty-aware methods is the ability to perceive consistent and complementary information between different views. Therefore, we conducted a set of comparative experiments with TMC in the Toy Dataset to investigate the effectiveness of CCML in solving semantic vagueness questions and to explicitly achieve higher accuracy results.

We set view 1 cannot distinguish categories $c_2$ and $c_3$ and view 2 cannot distinguish categories $c_1$ and $c_2$, respectively. Specifically, the toy dataset consists of 2 views, each with 1200 data instances $\{x_n^v\}_{n=1}^{1200}$ belonging to 3 categories, $c_1$, $c_2$, and $c_3$, with 400 data instances in each category. The underlying latent space has 9 dimensions, with three for each category. The first 3 dimensions and the last 3 dimensions are private to categories $c_1$ and $c_3$ respectively, and the middle dimensions are a shared dimension for $c_2$ and $c_3$. Each element of $\{v_n^v\}_{n=1}^{1200}$ is the sum of a number sampled from a gamma-distributed $\Gamma(1, 0.9)$, the noise sampled from Gaussian distribution $N(0, 0.1)$, and a consistent term of 0.5. We randomly generated $12 \times 9$ basis matrices $W^v$ for the two views, with elements drawn from a uniform distribution $U(0.4, 1)$, We randomly set 30 percent of the elements to be zero to simulate the real-world multi-view mapping pattern. Then we generated a noise matrix $Z^v$, and the elements of $Z^v$ drawn by the Gaussian distributions $N(0, 0.5)$ and $N(0, 1)$, respectively. We use the equation $x_n^v = W^v v_n^v + z^v$ to generate data instances.

In the Toy Dataset, we set the last 3 columns of $W^1$ to be 0 and the first 6 columns of $W^2$ to be 0, respectively. Therefore, the data instances in view 1 cannot distinguish between categories $c_2$ and $c_3$, and the data instances in view 2 cannot distinguish between categories $c_1$ and $c_2$. The Toy Dataset represents a strong

**Table 1: Classification accuracy (%) and uncertainty on the toy dataset.**

|      | Accuracy        | Uncertainty |
| ---- | --------------- | ----------- |
| TMC  | 94.73 ± 0.25    | 0.58        |
| CCML | 98.07 ± 0.28    | 0.23        |

complementarity between the two perspectives. To clarify, we use t-SNE to visualize the multi-view data instances of the Toy Dataset, as shown in Figure 3.

In addition, to obtain a better understanding of the advantages of CCML over TMC in addressing semantic vagueness problems, we have visualized the evidence produced by CCML and TMC for instances in category $c_2$, as illustrated in Figure 4.

Based on the experimental results, we can obtain the following conclusions: (1) The accuracy rate of CCML on the Toy Dataset is 98.07%, which is significantly higher than the accuracy rate of TMC on the same dataset, which is 94.73%. This indicates that CCML demonstrates superior performance in addressing semantic vagueness problems. (2) Upon observation in Figure 4, it is evident that when facing semantic vagueness problems, the evidence generated by a single view in the TMC method is subdued due to the inherent ambiguity of the view. Consequently, this results in insufficient evidence. The insufficiency arises because vague views fail to distinguish between certain categories, resulting in decision conflicts within the TMC framework, and ultimately, an inadequate evidence supply for both categories. In contrast, CCML can effectively generate sufficient evidence from each view and effectively leverage the information from vague views through the consistent evidence component. As a result, CCML constructs accurate and effective evidence supporting the $c_2$ category.

### 4.2 Experiment on Real-world Datasets

#### 4.2.1 Experimental Setup.

*Datasets.* **HandWritten**[1] comprises 2000 instances of handwritten numerals ranging from '0' to '9', with 200 patterns per class. It is represented using six feature sets. **Scene15**[2] includes 4485 images from 15 indoor and outdoor scene categories. We extract three types of features HOG, LBP, and GIST. **CUB** [30] consists of 11788 instances associated with text descriptions of 200 different categories of birds, we focus on the first 10 categories and extract image features using GoogleNet and corresponding text features using doc2vec. **LandUse** [38] comprises 2100 images from 21 classes. We extract HOG and SIFT features as two views. **PIE**[3] contains 680 facial instances belonging to 68 classes. We extract intensity, LBP, and Gabor as 3 views. **Colored-MNIST**[4] includes 1200 instances of numerals with RGB coloured backgrounds which consist of three colours (red, green, blue) for each number. We extract RGB and HOG features as two views.

---

[1]https://archive.ics.uci.edu/dataset/72/multiple+features
[2]https://figshare.com/articles/dataset/15-Scene_Image_Dataset/7007177/1
[3]http://www.cs.cmu.edu/afs/cs/project/PIE/MultiPie/Multi-Pie/Home.html
[4]https://www.kaggle.com/datasets/youssifhisham/colored-mnist-dataset/

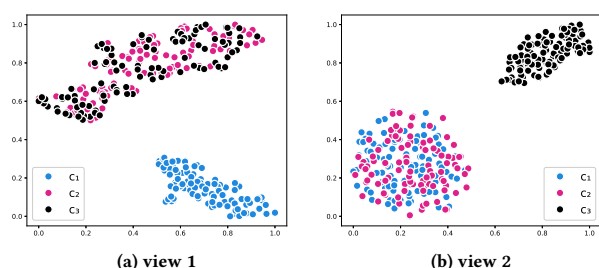

**(a) view 1**          **(b) view 2**

**Figure 3: Visualization of data instances in the toy dataset.**

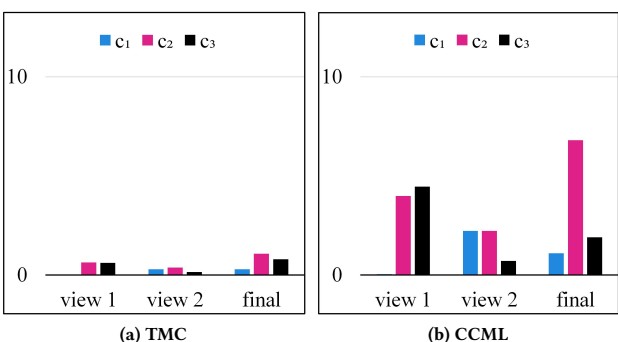

**(a) TMC**          **(b) CCML**

**Figure 4: The evidences of instances of $c_2$ on the toy dataset.**

*Compared Methods.* (1) **Single-view uncertainty aware methods** contain: EDL (Evidential Deep Learning) [27] and IEDL [3] which is the SOTA method that combines Fisher's information matrix. (2) **Multi-view feature fusion methods** contain: DCCAE (Deep Canonically Correlated AutoEncoders) [31] is the classical method, which employs autoencoders to seek a common representation, DCP (Dual Contrastive Prediction) [18] is the SOTA method that obtains a consistent representation. (3) **Multi-view decision fusion methods** contain: CALM (Enhanced Encoding and Confidence evaluating framework) [42] takes advantage of cross-view consistency and diversity to improve the efficacy of the learned latent representation, ETMC (Enhanced Trusted Multi-view Classification) [10], addresses the uncertainty estimation problem and produces reliable classification results. RCML (Reliable Conflictive Multiview Learning) [35] is the SOTA method that proposed a fusion strategy for solving conflictive problems.

*Implementation Details.* We briefly introduce the details of the experiment. We utilize fully connected networks with a ReLU layer to extract view-specific evidence. The Adam optimizer [16] is used to train the network, where L2-norm regularization is set to $1e^{-5}$. We employ 5-fold cross-validation to select the learning rate from the options of $3e^{-3}$, $1e^{-3}$, $3e^{-4}$, $1e^{-4}$. In all datasets, 20% of the instances are allocated as the test set. The average performance is reported by running each test case five times.

#### 4.2.2 Performance Comparison.
We compare CCML with the other classification methods, and the results are shown in Table 2. We can

**Table 2: Classification accuracy (%) on different datasets.**

| Data | EDL | IEDL | DCCAE | DCP | CAML | ETMC | RCML | CCML |
|---|---|---|---|---|---|---|---|---|
| Handwritten | 97.00 ± 0.16 | 98.45 ± 0.43 | 97.05 ± 0.24 | 88.10 ± 1.09 | 98.10 ± 0.12 | 98.32 ± 0.22 | 98.70 ± 0.19 | **99.28 ± 0.08** |
| Scene15 | 60.60 ± 0.13 | 65.40 ± 1.70 | 64.26 ± 0.42 | 72.08 ± 1.65 | 70.17 ± 0.13 | 66.87 ± 0.29 | 71.28 ± 0.32 | **74.76 ± 0.85** |
| CUB | 89.51 ± 0.24 | 92.67 ± 2.35 | 85.39 ± 1.36 | 93.00 ± 2.33 | 94.33 ± 0.73 | 90.81 ± 0.38 | 93.28 ± 2.75 | **94.58 ± 1.30** |
| LandUse | 47.10 ± 1.71 | 49.15 ± 0.28 | 50.42 ± 0.26 | 53.43 ± 3.67 | 58.18 ± 1.21 | 52.06 ± 0.71 | 53.55 ± 0.33 | **58.70 ± 1.75** |
| PIE | 87.99 ± 0.56 | 90.85 ± 3.31 | 81.96 ± 1.04 | 87.24 ± 2.48 | 93.38 ± 0.80 | 90.72 ± 0.21 | 93.89 ± 2.46 | **94.56 ± 1.83** |
| Colored-MNIST | 38.41 ± 0.43 | 44.83 ± 3.23 | 40.35 ± 0.67 | 87.15 ± .058 | 80.26 ± 0.39 | 83.76 ± 1.27 | 42.11 ± 2.01 | **91.54 ± 1.48** |

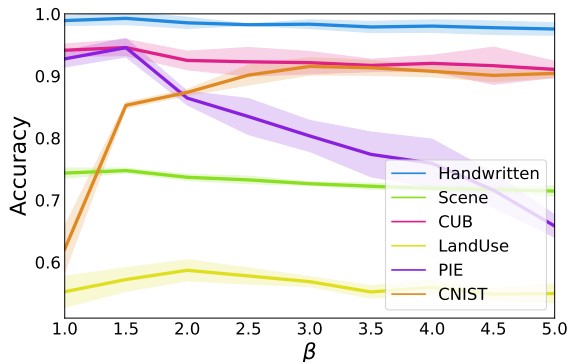

**Figure 5: The accuracy with different hyper-parameter $\beta$.**

obtain that: (1) multi-view methods generally outperform single-view methods, which illustrates the necessity of using multiple views in classification tasks. (2) There is a large difference in accuracy among different methods on the Colored-MNIST dataset, demonstrating significant variation in the ability of different methods to solve the semantic vagueness problem. (3) For the majority of real-world datasets, CCML shows performance comparable to state-of-the-art methods and has outstanding performance on Colored-MNIST datasets. (4) This result indicates that CCML significantly improves the ability to handle the semantic vagueness phenomenon while ensuring good performance on general classification tasks. The reason would be attributed to the consistent and complementary dynamic decoupling method, we would further verify this in the next analysis and other experiments.

### 4.2.3 Analysis.

*Ablation Study.* To validate the effectiveness of the evidence decoupling strategy, separability principle, and error evidence reduction principle, we construct a detailed ablation study that performs different combinations of these modules to achieve degradation methods. Specifically, to verify the effectiveness of the evidence decoupling module, we compared the approach with the three modules removed to using only that module. The effectiveness of the other two modules was validated based on the decoupling module respectively. We conduct the degradation methods above on the dataset Colored-MNIST. The results are shown in Table 3. From the result, we can observe the outstanding effectiveness of our dynamic decoupling strategy. Therefore, to further validate the effectiveness

**Table 3: The ablation study on the Colored-MNIST dataset.**

| Modules | | | ACC(%) |
|---|---|---|---|
| Decoupling | KL-divergence | Separation | Colored-MNIST |
| - | - | - | 57.50 ± 2.77 |
| ✓ | - | - | 80.37 ± 2.53 |
| ✓ | ✓ | - | 85.50 ± 2.10 |
| ✓ | - | ✓ | 89.79 ± 2.63 |
| ✓ | ✓ | ✓ | 91.54 ± 1.48 |

of the evidence decoupling module, we propose 6 variants of the CCML which Only use Consistent Evidence, Only Complementary Evidence, simple Accumulation of Evidence, simple Average of Evidence, Average of Evidence with Separation, and Accumulate of Evidence with Separation, respectively. We conduct CCML and variants on the CUB datasets and obtain experimental results as shown in Table 4. Compared to other methods, CCML achieves higher accuracy because it decouples and processes consistent and complementary evidence separately, giving higher confidence and increasing the separation degree for consistent evidence while averaging complementary evidence. This allows CCML to adjust the corresponding uncertainties based on the consistency between different views, instead of considering only one type of evidence or applying the same separation processing strategy to both consistent and complementary evidence. From the ablation study, we verified the effectiveness of each module of CCML.

*Parameter Analysis.* The separation increase module can improve the model's performance in solving classification tasks. We verify the influence of the separation increase module on the model by changing the value of the hyperparameter $\beta$. Specifically, we gradually increase the hyperparameter $\beta$ from 1 to 5 and observe CCML's performance on all datasets as shown in Figure 5. The results show that the accuracy of the model increases first and then decreases with the change of $\beta$. In particular, the effectiveness of $\beta$ in the semantic vagueness phenomenon is demonstrated on the dataset Colored-MNIST by the significantly increased accuracy. We can obtain the point that when the $\beta$ is too large, it will have a negative impact on the model, and the appropriate $\beta$ value can improve the performance of the model to a certain extent.

*Uncertainty Estimation.* To further evaluate the estimated uncertainty, we used the original dataset CUB and constructed out-of-distribution instances. We consider the original test instances as

Table 4: Comparison of CCML and variants on the CUB dataset.

| Data | CUB | | | | | | |
|---|---|---|---|---|---|---|---|
| Consistent Evidence | ✓ | - | ✓ | ✓ | ✓ | ✓ | ✓ |
| Complementary Evidence | - | ✓ | ✓ | ✓ | ✓ | ✓ | ✓ |
| Aggregate Strategy | - | - | Average | Accumulate | Average | Accumulate | CCML |
| Separation | - | - | - | - | ✓ | ✓ | ✓ |
| ACC(%) | 90.08 ± 0.80 | 74.46 ± 3.24 | 90.07 ± 0.32 | 91.54 ± 0.83 | 92.51 ± 0.32 | 92.19 ± 0.64 | 94.58 ± 0.24 |

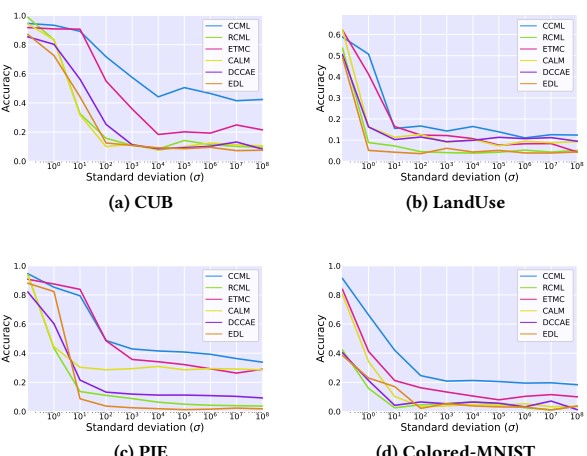

(a) CUB  (b) LandUse

(c) PIE  (d) Colored-MNIST

Figure 6: Comparison of all baseline on noise datasets.

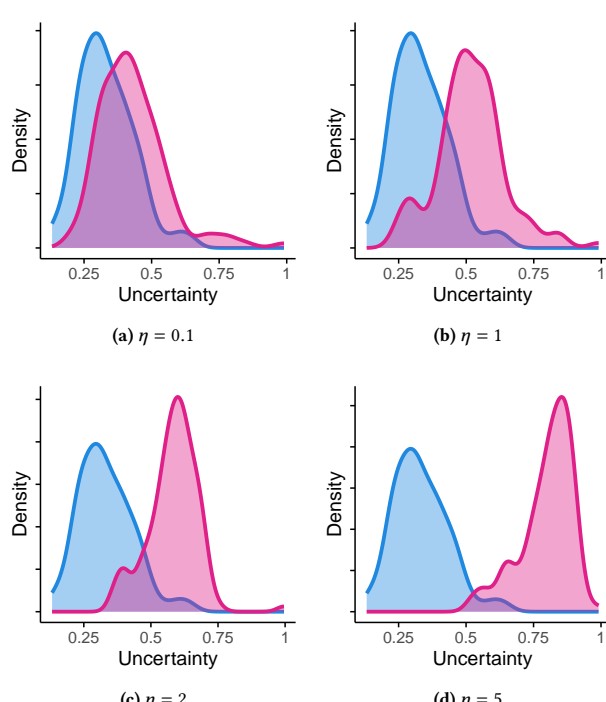

(a) $\eta = 0.1$  (b) $\eta = 1$

(c) $\eta = 2$  (d) $\eta = 5$

Figure 7: Uncertainty comparison on noise datasets.

in-distribution data and add Gaussian noise $N(0, I)$ with intensity $\eta$ to the test instances in one view, constructing out-of-distribution test instances. Specifically, given the noise vector $\epsilon$ sampled from the Gaussian distribution, the out-of-distribution test instances $\widetilde{x}_i = x_i + \eta \epsilon_i$. The uncertainty associated with out-of-distribution data is expected to be higher compared to that of in-distribution data. The noise intensity increases in the sequence ($\eta = 0.1, 1, 2, 5$). We perform CCML on the data with added Gaussian noise and visualize the uncertainty, as shown in Figure 7, where the blue curves represent in-distribution instances and the red curves represent out-of-distribution instances with noise intensity $\eta$. The results show that as the intensity of noise increases, the overall distribution of the uncertainty also increases, demonstrating that the data have higher uncertainty with greater noise. This also demonstrates the ability of our method to estimate uncertainty.

*Noise Views Impact Analysis.* To verify the robust classification ability of CCML for noise views, we construct noise instances and conduct CCML and other baseline methods. Specifically, we select about half of the views and add Gaussian noise with distribution $N(0, \sigma^2)$ to the instances of these noise views, where $\sigma$ is the standard deviation. The result is shown in Figure 6. From the experimental results, we can observe that as the standard deviation of Gaussian noise increases, the classification performance of all methods deteriorates. However, the deterioration in CCML's accuracy is noticeably smaller than that for other baseline methods in most

cases. The possible reason for this is that CCML can dynamically adjust its evidence obtained between different views. In the case of more noise within views, the consistency of evidence between viewpoints decreases, which prevents the training process from overly trusting the noise and thereby generating incorrect classification results.

## 5 CONCLUSION

In this paper, we propose a CCML method to solve the semantic vagueness problem in trusted multi-view learning. CCML tries to dynamically decouple the consistent and complementary evidence from the view-specific evidence. It further processes consistent and complementary evidence according to different principles to achieve classification results and reliability. The experimental results on the synthetic toy dataset and six real-world datasets verified the effectiveness of the proposed decouple strategy and the performance superiority of CCML.

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
