# OpenReview forum: "Dynamic Evidence Decoupling for Trusted Multi-view Learning"
_acmmm.org/ACMMM/2024/Conference — MM2024 Oral_

### Official Review · Reviewer_7gRU · 2024-05-03

**Rating:** 3
**Confidence:** 2

**Summary:**

This work proposes a trusted multi-view classification framework that separates consistent and complementary evidences based on the existence of semantic vagueness phenomenon in multi-view learning.

**Strengths:**

##  **Advantages.**
1. The motivation is clear and reasonable.
2. The method is easy to understand.
3. The experiments on simulated and real datasets show that the proposed method has good multi-view fusion effects.

**Limitations:**

## **Some obstacles encountered.**
However, after reading the entire paper and the code in the supplementary materials, I still encountered obstacles in understanding this paper.

### **1. The correctness of the code implementation.**

**In the code of model.py:** Let's put aside the confusing naming of variables, **(1)** In the function of **"def Evidence_DC"**, why is the **"alpha_con = E_con + 1"** added twice (after codes "E[v] = torch.sub(E[v], E_con)" and "alpha_a = E_a + 1")? Because according to the definition of ecidence theory [1-2], an evidence and its parameter can be presented as $\boldsymbol{\alpha}$=$\boldsymbol{e}$+1, as defined in this paper, too. It can be inferred that **$\boldsymbol{\alpha}^{con}$**=**$\boldsymbol{e}^{con}$**+1. But obviously, if the code is added with a **1** twice here, it clearly does not comply with the rules of evidence theory [1-2]. Therefore, I reasonably speculate that there is a certain technical error here.

**(2)** In addition, after the code "E_div = torch.div(E_div, len(alpha))", there are also some confusing codes appearing for readers. From the line "S_con = torch.sum(alpha_con, dim=1, keepdim=True)" to "# print(E_con[0])". Due to its chaotic naming, I reluctantly understood after carefully reading the code that this was to update Eq.(6) and obtain new evidence $\boldsymbol{e}^{con}$ and its parameters $\boldsymbol{\alpha}^{con}$. However, what confuses me the most are "Sum0_con = torch.sum(E_con, dim=1, keepdim=True)" and "Sum1_con = torch.sum(E_con, dim=1, keepdim=True)", and I understand that they are updating according to Eq.(6). Why do they count as the sum of categories of evidence? Shouldn't it be the sum of parameter categories? Because by definition in ecidence theory [1-2], $\boldsymbol{b}^{con}$ ($\tilde{\boldsymbol{b}^{con}}$ is the same idea) is related to Dirichlet intensity $S^{con}$ and $S^{con}$ is related to the sum of $\boldsymbol{\alpha}^{con}$ as $\sum_{k=1}^{K}\alpha_{k}^{con}$, rather than $\sum_{k=1}^{K} e^{con}_{k}$.

**These two points seriously affect the correctness and reproducibility of the proposed method implementation, and I hope the author can clarify my these two concerns.**

[1] Upper and lower probabilities induced by a multivalued mapping. The Annals of Mathematical Statistics 1967.

[2] Evidential deep learning to quantify classification uncertainty. NIPS 2018.

### **2. Some concerns about the motivations.**
**About the motivations:**  The authors used a bimodal application example in the abstract and introduction to introduce the challenge of semantic vagueness, which is intuitively reflected in Fig. 1. However, when I moved on to the experimental setup section, I found that the datasets used by the authors should be divided into two different types of multi-views: Modal-mixing (CUB) and feature-descriptor-mixing (HandWritten, Scene15, LandUse, PIE, Colored-MNIST). And the previous motivation sections only shows the modal-mixing case.
The phenomenon of semantic vagueness in the feature-descriptor-mixing case should also be explained with a simple example and added into Fig. 1.

**About the toy example:** In addition, the only visualization example in the experiment, Fig. 3, only shows the semantic vagueness challenge they try to simulate. But there is no display of the t-SNE effect after using the proposed method. The ideal effect after classification should be more distinguishable. This can reverse verify that the model indeed solves this challenge during multi-view fusion learning as they claim, but the authors did not further show it.

**In summary, there are still some concerns in this paper for me. If the authors can perfectly solve my confusion in code and other aspects of this paper, I am willing to change my score.**

**Suitability:**

3

---

### Official Review · Reviewer_znEU · 2024-05-03

**Rating:** 4
**Confidence:** 4

**Summary:**

## **1. Summary.**
This paper propose a Consistent and Complementary-aware trusted Multi-view Learning (CCML) method to solve the problem of semantic vagueness phenomenon in real-world multi-view data.

**Strengths:**

## **2. Strengths.**
i) This work is different from traditional uncertainty estimation fusion methods. It distinguishes evidence through the principles of consistency and complementarity, and designs corresponding losses. Thus, this approach has a certain degree of innovation.

ii) A certain degree of theoretical analysis provides theoretical support for the proposed framework.

iii) The experimental design is reasonable and rich.

**Limitations:**

## **3. Some details worth discussing.**
After carefully reading this paper, I have identified some limitations that can improve the quality of the paper and hope to discuss them with the authors.

**Contribution of this work:** The author mentioned "Decision fusion methods [5, 23, 26, 28] mitigate this problem by integrating the decision results from different views. We follow this line and propose a new method for decoupling consistent and complementary information at the decision level." in related work, the fact is indeed so. As for this paper, when we move our perspective to the loss section, we will find that it is very similar to the loss in work [1-2], but plays a different role. In other words, the contribution of the paper lies only in proposing an improved evidence theory fusion method based on the principles of consistency and complementarity. From this perspective, the contribution of this paper is incremental. But after reading, I have a clear understanding of the design concept of this paper, and it seems that the work is also easy for other readers to understand. Therefore, it is necessary to clarify the similarities and differences between this paper and other similar works, such as [1-2]. Therefore, I suggest the authors add a separate section to focus on discussing the similarities and differences between this work and other similar methods in terms of fusion models and losses.

[1] Trusted multi-view classification. ICLR 2021.

[2] Trusted multi-view classification with dynamic evidential fusion. TPAMI 2022.

**About theory:** I am pleased to see the author's efforts in theoretical support, including the proof in the Appendix. However, I also hope to see a theoretical analysis of the computational complexity of the proposed method. Furthermore, it would be even better if it could include an overall algorithmic process, which would be user-friendly for readers.

**About experiments:** **i)** Can the authors add some larger multi-view datasets for testing to show the performance under large-scale case? For example, NoisyMNIST (with 30,000 samples).
**ii)** Is it fair to only use accuracy (ACC) to reflect "trustworthiness" in trusted multi-view learning? The use of other indicators may bring progress to further trusted explanations, such as ECE, MCE [3].

[3] On calibration of modern neural networks. ICML 2017.

**Other details:** Some styles can be more unified to make reading more enjoyable for readers, such as subtitle "Experimental Setup" and "Implementation Details", they all can use command "\subsubsection" to form a unified style.
Some discussions on future work also need to be added. This can enable readers to clearly grasp the research trends of follow-up work.

**Overall, the ideas presented in this paper are somewhat novel, but there is still room for improvement. If authors can adopt some of the suggestions, I am willing to keep my rating.**

**Suitability:**

3

---

### Official Review · Reviewer_wokg · 2024-05-20

**Rating:** 6
**Confidence:** 4

**Summary:**

This paper demonstrates that the semantic vagueness phenomenon significantly suppresses the learning of view-specific evidence in existing methods. Then, they propose a CCML method to solve this problem by dynamically decoupling the consistent and complementary evidence. Experiment results validate the effectiveness of the dynamic evidence decoupling strategy and show that CCML significantly outperforms baselines on accuracy and reliability.

**Strengths:**

-This paper is well motivated. It points out a intersting problem, semantic vagueness phenomenon, in existing trusted multi-view learning methods. The authors also show
intuitive examples in figure 1 to explain this.

- The proposed method is convenient and effective. The proposed CCML dynamically decouples the consistent and complementary evidence, and allowing potential vagueness in the complementary evidence.

- This paper is well written and the experiment shows excellent performance.

**Limitations:**

- The authors claim that "the evidence aggregation paradigms in these methods
rely on an assumption: the data of each view can distinguish all categories." It is better to detailedly claim the reason behind this conclusion.

- This is only an advise. The proposed method is only suitable to classification  task?

**Suitability:**

3

---

### Official Review · Reviewer_VXxn · 2024-05-23

**Rating:** 2
**Confidence:** 3

**Summary:**

This paper proposes a trusted multi-view deep learning method through Dynamic Evidence Decoupling, which aims to mitigate the impact of the semantic vagueness phenomenon. Specifically, the method devises a consistent and complementary-aware multi-view fusion strategy to enhance the multi-view classification performance.

**Strengths:**

1. Trusted multi-view deep learning is an interesting topic.

2. The experimental evaluation has demonstrated good performance in terms of accuracy.

**Limitations:**

1. If I understand correctly, there appears to be a significant technical error: $\widetilde {{w^{con}}}$ cannot ensure that the sum of the $\widetilde {{b^{con}}}$ and the uncertainty $u^{con}$ equals 1 under a uniform prior $a^{con}=1$, which contradicts the principles of subjective logic.

2. The innovation of this paper is limited; several works [1-4] have been proposed to mitigate the impact of conflicting views through consistent constraint [1], dissonance uncertainty [3] and conflict fusion [2, 4]. However, there is no convincing evidence to demonstrate CCML's strengths compared to existing papers, especially the works of [2, 4].

3. Under the constraint of KL-divergence, which shrinks the incorrect evidence to zero and enhances the correct evidence, is it necessary to introduce a consistent loss function, or a complementary loss function? It would be better to provide a deeper empirical and theoretical analysis to validate the effectiveness of this method.

4. What role does uncertainty play in multi-view fusion? It seems that there is no improvement to enhance the reliability of the results, only to enhance the accuracy.

5. Why should the formulation of $e^{con}$ in Eq.(1a) be multiplied by $V$? How can consistent evidence be converted into consistent opinions? What is the meaning of $u^{con}$ compared to the overall $u$? How is $a^{con}$ defined? All of these questions should be clarified in the manuscript.

6. The experiment on toy samples should be compared with existing works [2,4] that address the same problem, in order to enhance the strength of this method, rather than focusing on the TMC.

[1] Trusted multiview deep learning with opinion aggregation. AAAI 2022.

[2] Uncertainty-Aware Multiview Deep Learning for Internet of Things Applications. TII 2023.

[3] Safe multi-view deep classification. AAAI 2023.

[4] Reliable Conflictive Multi-View Learning. AAAI 2024.

**Suitability:**

3

---

### Meta-Review · Area_Chair_qYvA · 2024-07-03

**Recommendation:** Accept (Oral)
**Confidence:** 4

**Metareview:**

This paper originally received mixed ratings, especially many comments on confusing formulations. However, after rebuttal, these questions have been resolved and all reviewers give a positive review. Thus AC recommends acceptance. Upon acceptance, it is recommended to include a discussion on tasks beyond classification.